# The impacts of social restrictions during the COVID-19 pandemic on the physical activity levels of over 50-year olds: The CHARIOT COVID-19 Rapid Response (CCRR) cohort study

Conall Green[1], Thomas Beaney[1]*, David Salman[1,2]*, Catherine Robb[3,4], Celeste A. de Jager Loots[4], Parthenia Giannakopoulou[4], Chi Udeh-Momoh[4], Sara Ahmadi-Abhari[5], Azeem Majeed[1,6], Lefkos T. Middleton[4,6], Alison H. McGregor[2]

1 Department of Primary Care and Public Health, Imperial College London, London, United Kingdom, 2 MSk Lab, Faculty of Medicine, Imperial College London, London, United Kingdom, 3 Epidemiology and Preventive Medicine Alfred Hospital, Monash University, Melbourne, Australia, 4 Ageing Epidemiology Research Unit (AGE), School of Public Health, Imperial College London, London, United Kingdom, 5 Department of Epidemiology and Biostatistics, School of Public Health, Imperial College London, London, United Kingdom, 6 Public Health Directorate, Imperial College Healthcare NHS Trust, London, United Kingdom

* thomas.beaney@imperial.ac.uk (TB); d.salman11@imperial.ac.uk (DS)

## Abstract

### Objectives

To quantify the associations between shielding status and loneliness at the start of the COVID-19 pandemic, and physical activity (PA) levels throughout the pandemic.

### Methods

Demographic, health and lifestyle characteristics of 7748 cognitively healthy adults aged >50, and living in London, were surveyed from April 2020 to March 2021. The International Physical Activity Questionnaire (IPAQ) short-form assessed PA before COVID-19 restrictions, and up to 6 times over 11 months. Linear mixed models investigated associations between shielding status and loneliness at the onset of the pandemic, with PA over time.

### Results

Participants who felt 'often lonely' at the outset of the pandemic completed an average of 522 and 547 fewer Metabolic Equivalent of Task (MET) minutes/week during the pandemic (95% CI: -809, -236, p<0.001) (95% CI: -818, -275, p<0.001) than those who felt 'never lonely' in univariable and multivariable models adjusted for demographic factors respectively. Those who felt 'sometimes lonely' completed 112 fewer MET minutes/week (95% CI: -219, -5, p = 0.041) than those who felt 'never lonely' following adjustment for demographic factors. Participants who were shielding at the outset of the pandemic completed an average of 352 fewer MET minutes/week during the pandemic than those who were not (95% CI: -432, -273; p<0.001) in univariable models and 228 fewer MET minutes/week (95% CI:

**Data Availability Statement:** Anonymised data relating to this project have been approved for dissemination by the Ageing Epidemiology Unit (AGE) and the Faculty of Medicine Data Management team at Imperial College London in August 2023. It is aimed that this dataset will accompany future datasets in a designated repository, and this is being currently established. Until this point, data will be made available by contacting the data controllers for this dataset at: parthenia.giannakopoulou13@imperial.ac.uk and/or e.mckeand@imperial.ac.uk. When the destination repository has been established, requests for data will be signposted accordingly.

**Funding:** Work towards this article was in part supported by the National Institute for Health Research (NIHR) Applied Research Collaboration Northwest London and Imperial Biomedical Research Centre (BRC). DS and TB are supported by Welcome / NIHR BRC fellowships. The views expressed in this publication are those of the authors and not necessarily those of the National Institute for Health Research or the Department of Health and Social Care. Imperial College London is the sponsor for the CCRR study, and has no influence on the direction or content of the work. There was no external financial funding for the study. The funders had no role in study design, data collection and analysis, decision to publish, or preparation of the manuscript.

**Competing interests:** I have read the journal's policy and the authors of this manuscript have the following competing interests: Sara Ahmadi-Abhari declares funding from EIT-health for a brain ageing PhD school programme, and is an unpaid advisor for small-sized chronic care management start-up (Medsien); Chi Udeh-Momoh declares: funding from a project grant funding consortia that included Janssen R&D, Gates Foundation, Merck and Takeda, a project grant from RoseTrees Foundation Trust and a project grant from Alzheimers Research UK; funding for a speaking engagement at the Lausanne IX workshop, an engagement at the Meeting of the Minds Neuroscience Conference, and was an invited speaker at the Reserve in Dementia Conference; is a scientific advisor at the Brain and Mind Institute, Aga Khan University, Nairobi; and is an unpaid executive committee member at Biofluids-based Biomarker Professional Interest Area for iSTAART, and a board of trustee member for the British Society for Neuroendocrinology; Lefkos T. Middleton reports research funding from Janssen, Novartis, Merck and Takeda, outside the submitted work and had unpaid leadership roles at the Clinical Trials in Alzheimer's Disease (CTAD) executive

-307, -150, p<0.001) following adjustment for demographic factors. No significant associations were found after further adjustment for health and lifestyle factors.

## Conclusions

Those shielding or lonely at pandemic onset were likely to have completed low levels of PA during the pandemic. These associations are influenced by co-morbidities and health status.

---

## 1. Introduction

### 1.1 Background/Rationale

Before the COVID-19 pandemic in 2019, between 60–70% of adults over 75 years in the UK were physically inactive, performing less than 30 minutes of moderate intensity physical activity per week [1, 2], and 60–75% were not physically active enough for good health as defined by World Health Organization (WHO) [3] and UK [4] guidelines. In March 2020, restrictions on social interaction were introduced in the UK to slow the transmission of COVID-19 [5]. Since then, a range of social restriction measures have been implemented and released (Fig 1) [6]. Throughout the pandemic those aged 70 years or older or with underlying health conditions defined as 'clinically extremely vulnerable', were advised to adhere to more stringent social restrictions, including shielding, where all social contact outside of the household was prohibited [7].

Social restrictions were needed to reduce the spread of the disease but there were concerns that they may have led to reduced physical activity (PA) during the pandemic in the long-term [8]. Social isolation can be associated with reduced physical activity, both for the general population [9–12] and for older adults [13–15]. Given the implementation of social restrictions during the pandemic, the imposed social isolation on a significant proportion of the population may have increased their risk of physical inactivity. PA has significant benefits across the spectrum of health [16], preventing cardiovascular disease [17, 18], cancers [19, 20], improving mental health [21–23] and providing other benefits from cognitive to bone health [4]. Therefore, those isolated or lonely during the COVID-19 pandemic may be at risk of poor health due to prolonged inactivity [24, 25].

### 1.2 Objectives

We hypothesised that older adults who were lonely or shielding at the outset of the pandemic would have decreased their levels of PA from the onset of the pandemic. To investigate this, we aimed to identify and quantify the associations between i) loneliness and ii) shielding status with PA throughout the course of the pandemic period when the survey was conducted. Thus, we aimed to provide insight into the wider health impacts of the pandemic, and to identify whether targeted PA promotion measures may be needed for particular groups of individuals during and after the pandemic.

## 2. Methods

### 2.1 Study design

This study was approved by the Imperial College Research and Ethics Committee (ICREC) and Joint Research Compliance Office (22/04/2020; 20IC5942). All participants were required

committee, WW FINGERS, and the European Consortium of Alzheimer's Disease; Celeste A. de Jager Loots received a 1-year research contract from the Foundations FINGERS Brain Health Institute, Sweden which contributed to her salary, and receives annual payments from the MCI and B Vitamin project from the University of Oxford, and has an unpaid advisory role membership at foodforthebrain.org; David Salman is funded by an Imperial College and National Institute of Health Research (NIHR) Biomedical Research Centre (BRC) fellowship, and is a paid executive board member for the Primary Care Rheumatology and Musculoskeletal Medicine society (PCRMM). All authors have completed the ICMJE uniform disclosure form at www.icmje.org/coi_disclosure. pdf. This does not alter our adherence to PLOS ONE policies on sharing data and materials.

to provide written, informed consent before taking part in the study. Participants were recruited from the Cognitive Health in Ageing Register of Investigational, Observational and Trial Studies (CHARIOT) [26]. The register is made up of over 40,000 cognitively healthy (no diagnosis of dementia) volunteers aged 50 or over, living in Greater London. All members of the CHARIOT register were invited to take part in this CHARIOT Covid-19 Rapid Response (CCRR) study. Of these, 7,748 participants accepted and completed an initial survey. Participants were invited to participate in up to five further surveys at 6-week and 3-month intervals (Fig 1). They were able to complete the initial survey at any point after receiving the questionnaire invitation. However, they were required to complete follow-up surveys within 10 days of receiving the invite.

The initial survey contained 122 questions on demographics, diet, alcohol and smoking status, symptoms of COVID-19, functional activities, sleep, frailty, mental health, and PA (S1 File). PA was assessed using the International Physical Activity Questionnaire short-form (IPAQ) containing seven questions [27]. Participants were asked to complete this questionnaire in each of the survey waves, providing an estimate for their weekly activity one week before completing the survey. As part of the first survey, participants were also asked to recall their PA habits before the implementation of COVID-19 restrictions in March 2020 (pre-pandemic PA). Participants were asked how many days they spent completing any vigorous or moderate physical activity or walking in the previous week. They were then asked how much time they usually spend doing each of these activities.

Loneliness was assessed by the question from the Imperial College Sleep Quality questionnaire: "During the last month, have you experienced loneliness (felt isolated, with no companions)?". They were given the following options: "never", "rarely", "sometimes", or "often". Shielding status was assessed by asking participants "Are you currently shielding as per government guidelines for clinically extremely vulnerable groups?", with options of "Yes" or "No" (S1 File).

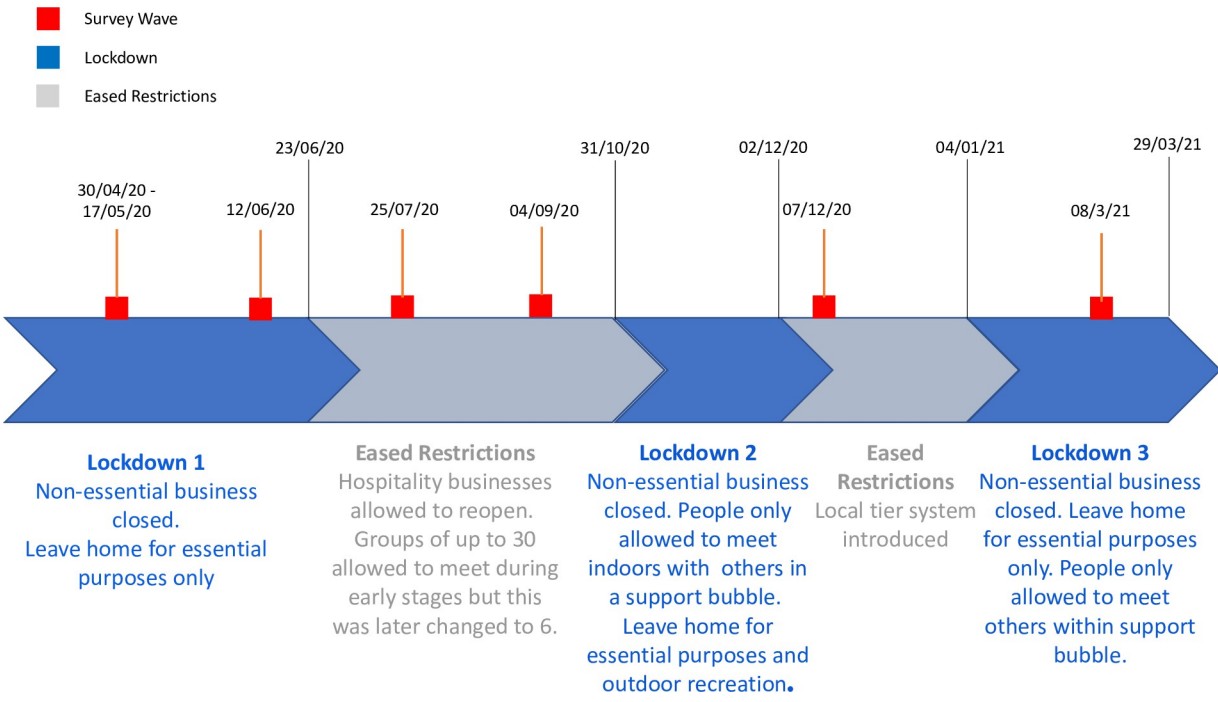

**Fig 1. A timeline of COVID-19 restrictions in the UK with the survey waves used for the study.**

## 2.2 Statistical methods

All statistical analyses were completed using R software version 4.0.2. The *lme4* package was used to create linear mixed models [28] and *lmerTest* to perform model validity tests [29]. Forest plots were produced with *Metafor* in R [30].

Body Mass Index (BMI) was calculated by dividing weight in kilograms by height in meters squared. IPAQ data were cleaned in accordance with IPAQ protocols [27]. Weekly Metabolic Equivalent of Task (MET) minutes, which represent the number of minutes at a certain intensity of energy expenditure per week (as multiples of resting metabolic rate), were calculated for each participant at each survey wave, as well as for activity levels before the implementation of restrictions (pre-pandemic PA). MET minutes were calculated by multiplying the following MET score values as defined and averaged by the IPAQ scoring protocol (walking = 3.3 METs, moderate PA = 4.0 METs, and vigorous PA = 8.0 METs) by the number of minutes completing the activity. For example, walking at a moderate pace for 5 minutes would represent 16.5 MET minutes [31].

The study investigates between-person differences in PA during the pandemic, adjusted for an individual's pre-pandemic PA. Two-level univariable linear mixed models were used, incorporating random intercepts for each participant, to assess the associations between shielding status, and loneliness at the point of the first survey, and time-varying PA at each survey. These models assumed equal slopes but allowed for different intercepts for each participant. A theoretical approach was used for confounder selection, aided by construction of two causal diagrams, for each exposure (S2.1 & S2.2 Figs in S2 File). The first multivariable model was adjusted for age, sex, ethnicity, month of survey completion (to account for possible seasonal effects) and pre-pandemic PA (Model 1). The second multivariable model was additionally adjusted for BMI and the presence or absence of one or more health conditions (Model 2). The third and final multivariable model was additionally adjusted for smoking (yes/no), alcohol consumption (yes/no), whether the participant was living alone (yes/ no) and whether the participant was single or in a relationship (Model 3). Equations for each model are shown in S3 File. Values of covariates were determined at the time of the initial survey, and the outcome measure of PA was time-updated at each survey wave. Statistical analyses and reporting aligns with the Checklist for statistical Assessment of Medical Papers (CHAMP) statement [32].

## 3. Results

### 3.1 Participant characteristics

Of the ~ 40,000 individuals on the CHARIOT register, 7748 consented to take part in the study. Those completing each survey ranged from 7748 (survey wave 1) to 4000 (survey wave 6) (Table 1). Participant characteristics are given in Table 2. Of the 7,748 participants included in the analysis from survey wave 1, 53.1% (4111) of the participants were female and the median age was 70 years, with a lower quartile of 66 years and upper quartile of 75 years. 89% of participants were of white ethnic background, 3.1% were of Asian background, 1.5% were of mixed or multiple ethnic origins, 0.7% were black African, Caribbean, or black British, and 1.1% were of other ethnicities. BMI data were missing for 66.7% of participants. The median BMI was 24.5 kg/m$^2$, with an interquartile range of 5.1 kg/m$^2$.

**Table 1. Number of participants completing each survey.**

| Survey | 1 (30/4/20) | 2 (12/6/20) | 3 (25/07/20) | 4 (04/09/20) | 5 (07/12/20) | 6 (08/03/21) |
|---|---|---|---|---|---|---|
| Number of completions | 7748 | 4884 | 4649 | 4725 | 4249 | 4000 |

**Table 2. Characteristics for 7748 participants at the point of the first survey; BMI–Body Mass Index; MET–Metabolic Equivalent of Task.**

| Characteristic | | N (%) |
|---|---|---|
| **Sex** | Male | 3297 (42.6%) |
| | Female | 4111 (53.1%) |
| | Missing | 340 (4.4%) |
| **Age (years)** | 50–59 | 808 (10.4%) |
| | 60–69 | 2592 (33.5%) |
| | 70–79 | 3553 (45.9%) |
| | 80–89 | 713 (9.2%) |
| | 90+ | 20 (0.3%) |
| | Missing | 62 (0.8%) |
| **Ethnicity** | White | 6896 (89.0%) |
| | Asian | 240 (3.1%) |
| | Black African, Caribbean or Black British | 54 (0.7%) |
| | Mixed or Multiple Ethnic Groups | 120 (1.5%) |
| | Other Ethnic Groups | 84 (1.1%) |
| | Missing | 354 (4.6%) |
| **BMI (kg/m$^2$)** | <18.5 | 51 (0.7%) |
| | 18.5–24.9 | 1382 (17.8%) |
| | 25–29.9 | 832 (10.7%) |
| | ≥30 | 316 (4.1%) |
| | Missing | 5167 (66.7%) |
| **Health Conditions** | Present | 4412 (56.9%) |
| | Absent | 3016 (38.9%) |
| | Missing | 320 (4.1%) |
| **Alcohol Drinker** | Yes | 5934 (76.6%) |
| | No | 1388 (17.9%) |
| | Missing | 426 (5.5%) |
| **Smoker** | Yes | 243 (3.1%) |
| | No | 7072 (91.3%) |
| | Missing | 433 (5.6%) |
| **Relationship Status** | Single | 2398 (30.9%) |
| | In a Relationship | 4928 (63.6%) |
| | Missing | 422 (5.4%) |
| **Loneliness** | Never | 3394 (43.8%) |
| | Rarely | 1660 (21.4%) |
| | Sometimes | 1484 (19.2%) |
| | Often | 473 (6.1%) |
| | Missing | 737 (9.5%) |
| **Shielding** | Yes | 2012 (26.0%) |
| | No | 5314 (68.6%) |
| | Missing | 422 (5.4%) |
| **MET minutes/ week** | ≤1000 | 1375 (17.7%) |
| | 1001–1500 | 974 (12.6%) |
| | 15001–2000 | 628 (8.1%) |
| | 2001–2500 | 645 (8.3%) |
| | 2501–3000 | 741 (9.6%) |
| | 3001–3500 | 446 (5.8%) |
| | >3500 | 1604 (20.7%) |
| | Missing | 1335 (17.2%) |

Of the population, 3.1% of participants were current smokers and 76.6% drank alcohol on a regular basis. Before restrictions were implemented, PA as measured by median MET minutes for participants was 1836 MET minutes/ week, with an upper quartile of 3252 MET minutes/ week and lower quartile of 816.5 MET minutes/ week.

The majority of participants reported being in a relationship (63.6%). A quarter of participants (26.0%) were shielding at the time of the first survey. At the start of the study 43.8% of participants reported never feeling lonely, 21.4% rarely being lonely, 19.2% sometimes feeling lonely and 6.1% often felt lonely. This question was left unanswered by 9.5% of participants.

### 3.2 Loneliness and physical activity

Results for all associations can be found in S4 File. In univariable linear mixed models, those who were often lonely completed an average of 522 fewer MET minutes/ week than those who were never lonely (95% CI: -809, -236, p<0.001). No significant difference was found between those who were rarely and never lonely in the univariable model (95% CI: -83, 80, p<0.968) (Fig 2).

After adjustment for age, sex, ethnicity, month of survey completion and pre-pandemic PA (model 1) those who were often lonely completed an average of 547 fewer MET min per week than those who were never lonely (95% CI: -818, -275, p<0.001). Those who were sometimes lonely completed an average of 112 fewer MET minutes/ week than those who were never lonely (95% CI: -219, -5, p = 0.041).

No significant differences were found between any of the levels of loneliness and never being lonely after additional adjustment for BMI and underlying conditions (model 2) or additionally adjusting for smoking status, alcohol consumption, whether the participant was living alone and whether the participant was single or in a relationship (model 3).

### 3.3 Shielding status and PA

In the univariable model those who were shielding at the onset of COVID-19 restrictions were found to complete an average of 352 fewer MET minutes/ week than those who were not (95% CI: -432, -273, p<0.001) (Fig 3).

After adjustment for age, sex, ethnicity, month of survey completion and pre-pandemic PA (model 1) those who were shielding completed an average of 228 fewer MET minutes/ week that those who were not (95% CI: -307, -150, p<0.001).

No significant difference in MET minutes/ week was found between those who were and were not shielding at the onset of COVID-19 restrictions after adjusting for age, sex, ethnicity, BMI, underlying conditions, month of survey completion and pre-pandemic PA (model 2) or after additionally adjusting for smoking status, whether the participant was an alcohol drinker, whether the participant was living alone and whether the participant was single or in a relationship (model 3) (Fig 3). Numbers of participants and observations, as well as coefficients for covariates for each model can be found in S4 File.

## 4. Discussion

### 4.1 Key results

We assessed the associations between measures of loneliness and shielding status at the start of the study, with long-term PA across the COVID-19 pandemic in adults from the CHARIOT cohort aged 50 years or over enrolled in the CCRR study. CCRR Study participants who were "often lonely" and those who were shielding at the onset of COVID-19 were significantly less

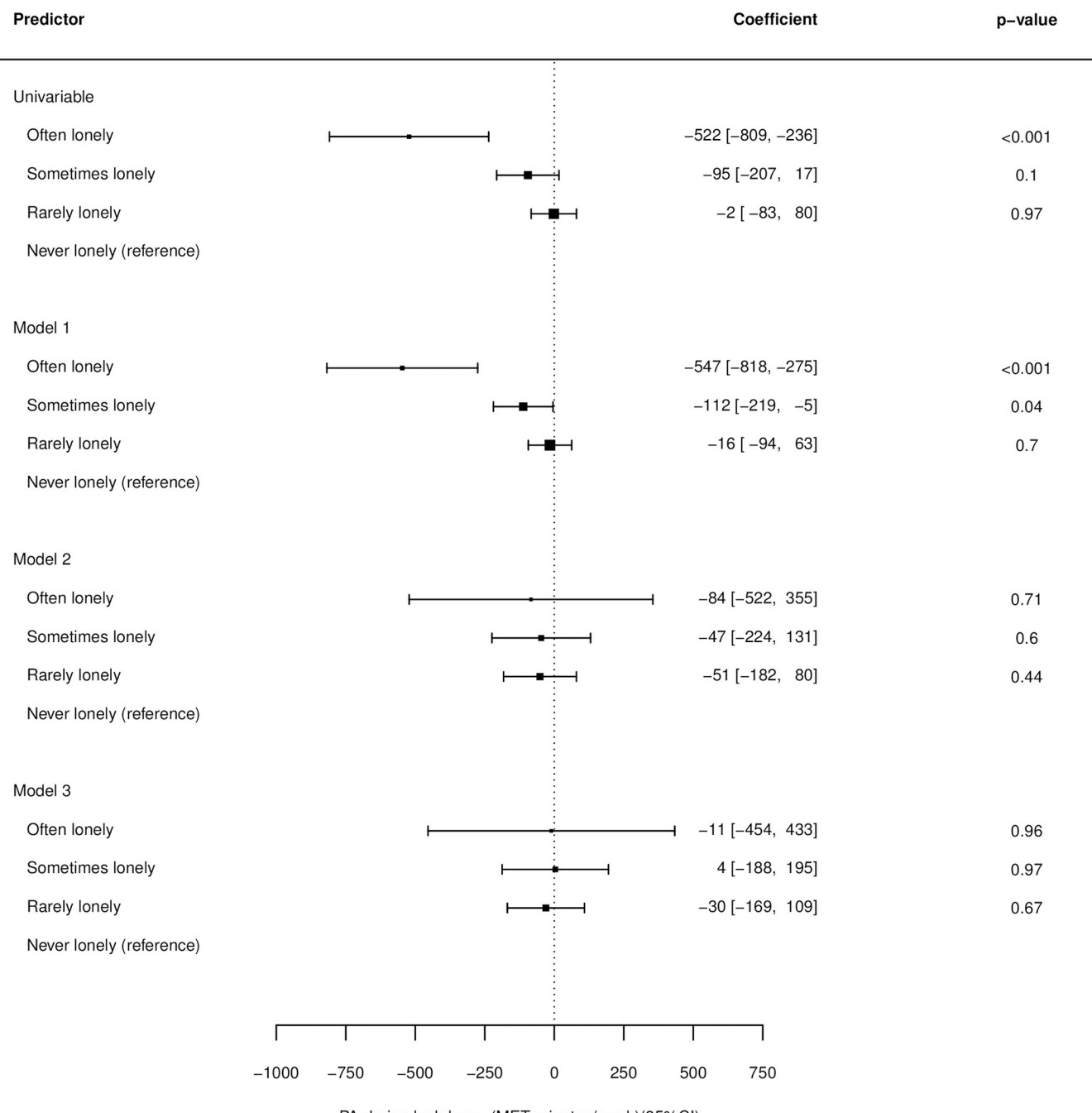

**Fig 2. Associations between loneliness and physical activity levels for the univariable model, model 1(adjusted for age, sex, ethnicity, month of survey completion and pre-pandemic Physical Activity—PA), model 2 (adjusted for age, sex, ethnicity, Body Mass Index—BMI, underlying conditions, month of survey completion and pre-pandemic PA) and model 3 (adjusted for age, sex, ethnicity, BMI, underlying conditions, month of survey completion, pre-pandemic PA, for smoking (yes/ no), whether the participant was an alcohol drinker (yes/ no), whether the participant was living alone (yes/ no) and whether the participant was single or in a relationship.**

physically active during the pandemic than those who were never lonely or not shielding. However, after adjustment for health and lifestyle factors, there was no significant association between loneliness or shielding status with PA.

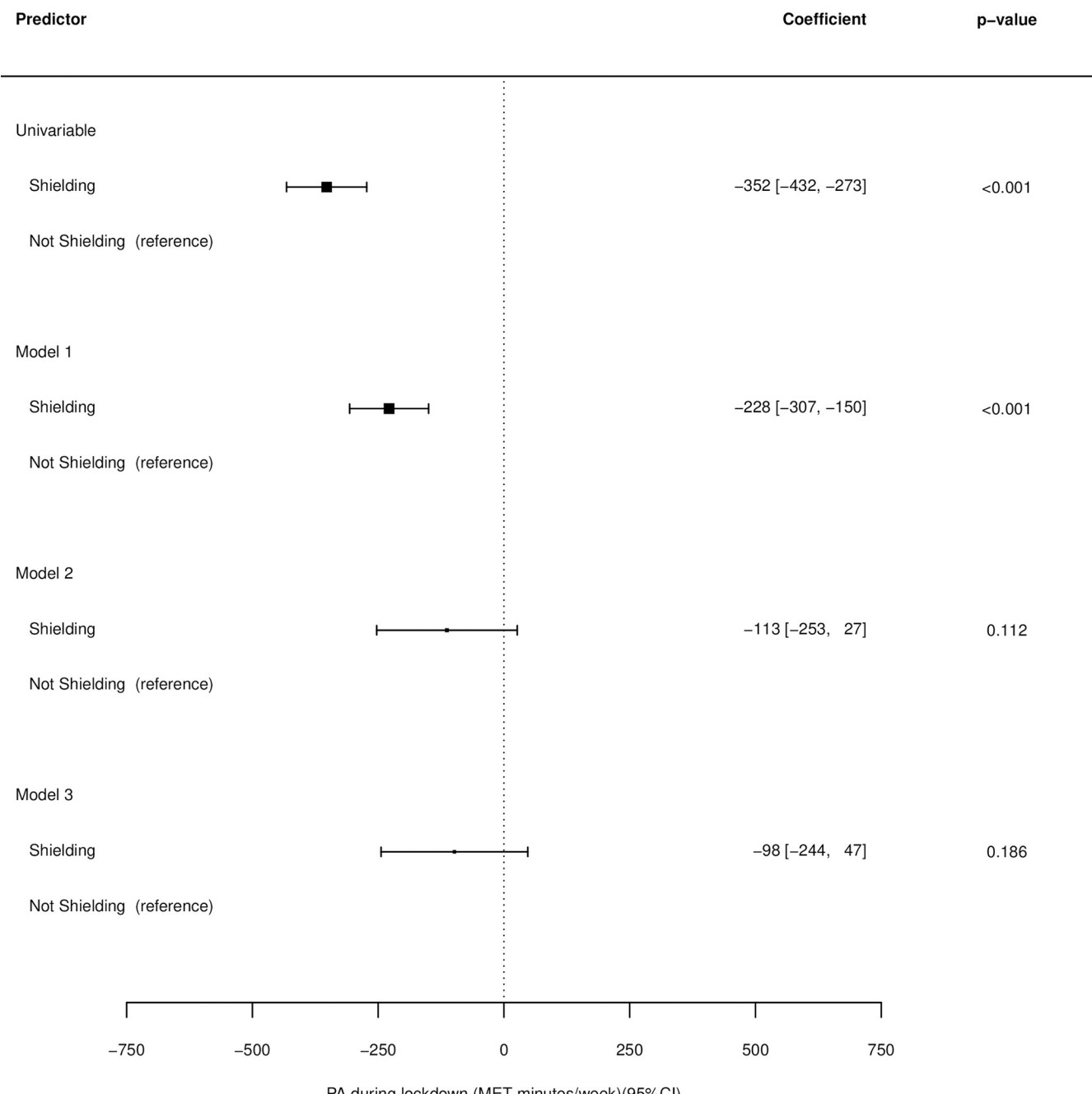

**Fig 3. Associations between shielding status and physical activity levels for the univariable model, model 1(adjusted for age, sex, ethnicity, month of survey completion and pre-pandemic Physical Activity—PA), model 2 (adjusted for age, sex, ethnicity, Body Mass Index—BMI, underlying conditions, month of survey completion and pre-pandemic PA) and model 3 (adjusted for age, sex, ethnicity, BMI, underlying conditions (present/ absent), month of survey completion, pre-pandemic PA, for smoking (yes/ no), whether the participant was an alcohol drinker (yes/ no), whether the participant was living alone (yes/ no) and whether the participant was single or in a relationship.**

## 4.2 Social isolation, loneliness and PA

Those who were "often lonely" at the onset of COVID-19 restrictions completed significantly less PA per week over the 11-month follow-up period than those who were never lonely. Similarly, those who were shielding completed significantly less PA than those who were not

shielding. Given that WHO minimum recommended PA guidance approximated 600 MET minutes per week [3], these differences found in those identifying as lonely or shielding represent a significant proportion of weekly PA. This aligns with previous evidence finding subjective and objective reductions in PA in those who are socially isolated or lonely [13, 33]. However, after further adjustment for health, lifestyle and relationship factors, including BMI, the presence of health conditions and living/relationship status, associations between PA and loneliness/shielding were no longer significant, indicating a confounding effect of these factors. This is possible as those of poorer health are likely to have been required to shield or isolate and complete lower levels of PA.

The mechanism for associations between social isolation or loneliness, and reduced PA is complex, and are likely related to the protective effect that social relationships have on health. For example, in a study of UK Biobank participants, negative health behaviours were a significant contributor to excess mortality in those who were socially isolated or lonely [34]. The relationship with physical activity might be due, in part, to the absence of motivation from external contacts [11], and reduced availability of social opportunities together with reduced capacity for PA. Loneliness is independently associated with reduced physical function [35]. These factors provide insights, and potential avenues for future interventions, in accordance with models of behaviour change [36] addressing motivation, opportunity and capability, respectively.

People who are socially isolated or lonely are likely to have other co-existing risk factors for lower PA. In a study of objective PA in the English Longitudinal Study of Ageing (ELSA), although social isolation was strongly associated with reduced PA and increased sedentary time in adults over 50 years of age even after adjustment for covariates, the association of loneliness with reduced PA was only present in univariable models [13]. Social isolation (an objective measure of social contacts) and loneliness (the subjective feeling of the difference between preferred and actual social contact) encapsulate different [37], but related, concepts. For PA, social connectedness may be a key driver, rather than the subjective perception of social contact alone.

Individuals who were often lonely or shielding at the outset of the pandemic may be at risk of poor health due to prolonged physical inactivity. These individuals may also be at risk of poor health due to other health consequences of loneliness and social isolation, such as poor mental health, cognitive impairment, and impaired motor function [38]. Studies have also identified associations between marital status, time alone, and mental or physical health and loneliness among older adults [39]. It is therefore possible that loneliness frequency increased during the pandemic due to reduced social contact, poorer physical health due to inactivity and poorer mental health due to the stress of the pandemic. Therefore, the negative health consequences of the pandemic might have longer-term effects and impacts beyond morbidity and mortality directly attributable to COVID-19.

## 4.3 Limitations

There are a number of limitations that may affect the generalisability of these results. First, although the IPAQ short-form is well validated across diverse populations under the age of 65 [40] and adequately validated in participants over this age [41], there may be bias in self-reported PA. Findings have shown that self-reporting tools only weakly correlate with objective measures such as pedometers [42–44]. Therefore, there may be some inaccuracy in the recording of PA.

There is a risk of recall bias within the study, as participants in the first survey wave were asked to recall their PA habits before the implementation of restrictions. These may have been over- or under-estimated. Although systematic differences in a participant recording higher or

lower estimates of their PA may be reduced by our assessment over long-term PA levels, there is likely to remain a bias in estimates at each time point. The methodology of this study may have been improved through the use of accelerometry and momentary social contact ratings on electronic diaries [45, 46]. These devices provide near real-time data assessment, reducing the risk of recall bias. Using the devices may have also improved the reliability of the study through allowing multiple assessments across time. However, given the additional financial cost of the devices and logistical challenges with using them among a large cohort throughout the pandemic, they were not used for this study. Similarly, the surveys were completed under varying social restrictions. Our study investigated the longer-term changes in PA from COVID-19 restrictions, but further work could explore the impact of external events, such as changes in social restrictions, on short-term changes in PA over the pandemic.

The questionnaire used a question on loneliness modified from the Imperial College Sleep Quality Questionnaire; which in turn was adapted from the Pittsburgh Sleep Quality Index [47] and Centre for Epidemiologic Studies of Depression Scale for work- free periods [48]. However, the Office for National Statistics (ONS) suggested measures of the first three questions from the University of California, Los Angeles (UCLA) three-item loneliness scale, together with a direct question about loneliness from the Community Life Survey, may add more validity or sensitivity to these surveys [49]. Moreover, the different categories used for loneliness are subjectively defined by the participant, potentially leading to instances where two individuals experiencing the same degree of loneliness may record two different answers. However, these limitations are largely due to the fact that loneliness is subjective, and cannot be objectively captured, making these issues largely unavoidable.

Another limitation of this study was the large amount of missing data for BMI, as many participants did not complete both the height and weight responses to the initial survey. This high level of missing data could impact on the level of significance of results for both loneliness and shielding multivariable models.

The criteria for shielding also changed throughout the pandemic [50]. Therefore, the effect of shielding on PA may vary between different risk groups. This makes the clinical meaning of the shielding results difficult to interpret. Finally, 89.0% of participants were white, whereas in the 2011 census this figure was much lower (44.9% of London's population identified as white British) [51]. Therefore, it is unlikely that the study participants are a true representation of London's population. Additionally, and perhaps related to this, the participants within the study were more active than expected. Over 90% of participants achieved the WHO's [3] recommended guidelines for PA pre-pandemic. This is significantly more than the 67% of London's population known to meet the recommended guidelines [52].

## 4.4 Conclusions

Participants who reported feeling often lonely or were shielding at the outset of the pandemic were found to be significantly less physically active during the 11-month study period compared to those who were never lonely or not shielding. Although these relationships were partially explained by other factors such as BMI, underlying conditions, and relationship status, the results indicate that people who were lonely or shielding may be at risk of poor health due to prolonged physical inactivity. Our findings highlight the need for proactive support for those experiencing loneliness or those shielding during the pandemic. Findings from this study also illustrate the need to support members of society who are socially isolated, or at high risk of loneliness outside of the context of the pandemic. Social isolation and loneliness should be considered in the design and implementation of physical activity interventions, and vice-versa, for these groups.

## Supporting information

**S1 Checklist. STROBE statement—checklist of items that should be included in reports of observational studies.**
(DOCX)

**S1 File. Variables extracted from survey.**
(DOCX)

**S2 File. Fig S2.1 causal diagrams for loneliness; Fig S2.2 causal diagrams for shielding.**
(DOCX)

**S3 File. Model equations.**
(DOCX)

**S4 File. Coefficients of study variables.**
(DOCX)

## Acknowledgments

We are grateful to Lesley Williamson, Monica Munoz-Troncoso, Snehal Pandya and Emily Pickering (CHARIOT register and facilitator team); Mariam Jiwani, Rachel Veeravalli, Islam Saiful, Danielle Rose, Susie Gold, Rachel Nejade and Shehla Shamsuddin (Imperial College London student volunteers); Stefan McGinn-Summers, Neil Beckford, Inthushaa Indrakumar and Kristina Lakey (Departmental administrative staff in AGE); Dinithi Perera (departmental manager); Heather McLellan-Young (project manager); Helen Ward, James McKeand, Geraint Price, Josip Car, Christina Atchison, Nicholas Peters, Aldo Faisal, and Jennifer Quint (investigator team contributing to CCRR survey design, development and improvement).

## Author Contributions

**Conceptualization:** Thomas Beaney, David Salman, Catherine Robb, Chi Udeh-Momoh, Sara Ahmadi-Abhari, Azeem Majeed, Lefkos T. Middleton, Alison H. McGregor.

**Data curation:** Conall Green, Catherine Robb, Parthenia Giannakopoulou, Chi Udeh-Momoh, Sara Ahmadi-Abhari.

**Formal analysis:** Conall Green, Thomas Beaney, Celeste A. de Jager Loots, Parthenia Giannakopoulou, Chi Udeh-Momoh, Sara Ahmadi-Abhari.

**Investigation:** Conall Green, Thomas Beaney, David Salman, Catherine Robb.

**Methodology:** Thomas Beaney, David Salman, Catherine Robb, Celeste A. de Jager Loots, Parthenia Giannakopoulou, Chi Udeh-Momoh, Sara Ahmadi-Abhari.

**Project administration:** Thomas Beaney, Catherine Robb, Celeste A. de Jager Loots, Parthenia Giannakopoulou, Chi Udeh-Momoh, Sara Ahmadi-Abhari.

**Resources:** Celeste A. de Jager Loots, Parthenia Giannakopoulou, Chi Udeh-Momoh, Sara Ahmadi-Abhari.

**Supervision:** Thomas Beaney, David Salman, Catherine Robb, Azeem Majeed, Lefkos T. Middleton, Alison H. McGregor.

**Writing – original draft:** Conall Green, Thomas Beaney, David Salman.

**Writing – review & editing:** Conall Green, Thomas Beaney, David Salman, Catherine Robb, Celeste A. de Jager Loots, Parthenia Giannakopoulou, Chi Udeh-Momoh, Sara Ahmadi-Abhari, Azeem Majeed, Lefkos T. Middleton, Alison H. McGregor.

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
