## [Decision Letter · Decision Letter 0]

29 Mar 2023

PONE-D-23-02121The impacts of social restrictions during the COVID-19 pandemic on the physical activity levels of over 50-year olds: the CHARIOT COVID-19 Rapid Response (CCRR) cohort studyPLOS ONE

Dear Dr. Salman,

Thank you for submitting your manuscript to PLOS ONE. After careful consideration, we feel that it has merit but does not fully meet PLOS ONE’s publication criteria as it currently stands. Therefore, we invite you to submit a revised version of the manuscript that addresses the points raised during the review process.

We look forward to receiving your revised manuscript.

Kind regards,

Julian Packheiser

Academic Editor

PLOS ONE

“I have read the journal's policy and the authors of this manuscript have the following competing interests: Sara Ahmadi-Abhari declares funding from EIT-health for a brain ageing PhD school programme, and is an unpaid advisor for small-sized chronic care management start-up (Medsien);  Chi Udeh-Momoh declares: funding from a project grant funding consortia that included Janssen R&D, Gates Foundation, Merck and Takeda, a project grant from RoseTrees Foundation Trust and a project grant from Alzheimers Research UK; funding for a speaking engagement at the Lausanne IX workshop, an engagement at the Meeting of the Minds Neuroscience Conference, and was an invited speaker at the Reserve in Dementia Conference; is a scientific advisor at the Brain and Mind Institute, Aga Khan University, Nairobi; and is an unpaid executive committee member at Biofluids-based Biomarker Professional Interest Area for iSTAART, and a board of trustee member for the British Society for Neuroendocrinology; Lefkos T. Middleton reports research funding from Janssen, Novartis, Merck and Takeda, outside the submitted work and had unpaid leadership roles at the Clinical Trials in Alzheimer’s Disease (CTAD) executive committee, WW FINGERS, and the European Consortium of Alzheimer’s Disease; Celeste A. de Jager Loots received a 1-year research contract from the Foundations FINGERS Brain Health Institute, Sweden which contributed to her salary, and receives annual payments from the MCI and B Vitamin project from the University of Oxford, and has an unpaid advisory role membership at foodforthebrain.org; David Salman is funded by an Imperial College and National Institute of Health Research (NIHR) Biomedical Research Centre (BRC) fellowship, and is an unpaid advisory board member for the Primary Care Rheumatology and Musculoskeletal Medicine society (PCRMM). All authors have have completed the ICMJE uniform disclosure form at www.icmje.org/coi_disclosure.pdf.”

Additional Editor Comments:

Dear Dr. Salman,

I have now received two expert reviews for your submitted manuscript. Both reviewers commend the large sample size but also raise significant concerns that should be addressed. I fully agree with their assessment that more information should be provided on how MET was calculated and that the models right now are not precisely described. An explicit model description providing the actual lmer specifications is necessary for the readers to understand what was calculated. While reading the manuscript, I also noted that your assessment of loneliness is rather unusual by asking one month in retrospect how participants rated their loneliness. A more common approach would be to use a dedicated questionnaire like the UCLA loneliness scale which provides a numeric and metric scale. Your questionnaire has ordinal properties as the difference between sometimes and never might not be equal compared to sometimes and often etc. This makes a linear model a strong assumption given that the predictor might not be linear and could represent an exponential increase in loneliness. The authors should consider using generalized linear mixed models with different specifications to check if treating loneliness as a linear predictor provides the best model fit. This kind of assessment should also be listed as a limitation. Furthermore, as reviewer 2 notes, high power also requires qualification of significant vs. meaningful. The authors could use Bayesian models using the brms and bayestestR packages to check for evidence in favor or against the null hypothesis in addition to frequentist statistics.

Reviewers' comments:

Reviewer's Responses to Questions

**Comments to the Author**

1. Is the manuscript technically sound, and do the data support the conclusions?

Reviewer #1: Partly

Reviewer #2: Partly

2. Has the statistical analysis been performed appropriately and rigorously? 

Reviewer #1: Yes

Reviewer #2: I Don't Know

3. Have the authors made all data underlying the findings in their manuscript fully available?

Reviewer #1: No

Reviewer #2: No

4. Is the manuscript presented in an intelligible fashion and written in standard English?

Reviewer #1: Yes

Reviewer #2: Yes

5. Review Comments to the Author

Reviewer #1: The authors investigate the associations of self-reported loneliness and shielding at the beginning of the COVID-19 pandemic and self-reported physical activity (PA) in the course of the pandemic in a large sample of cognitively healthy adults 50 years and older in the UK. Participants were grouped according to loneliness and shielding at the beginning of the study, and linear mixed effects models were employed to investigate the associations of these predictors with MET minutes, the employed physical activity measure. Results indicate that participants feeling lonely often and/or were shielding at baseline showed less MET minutes than those feeling never lonely or not shielding. These associations diminished when including health and lifestyle factors, such as BMI, health conditions, and relationship status. With their paper, especially due to the large sample size, the authors make a scientific contribution to understanding the associations of physical (in)activity and loneliness/social isolation in (older) adults, which might inform future prevention targets.

Still, I have some comments and suggestions for the authors:

Major comments:

- There seems to be a mismatch of the number of surveys depicted in Fig. 1, Table 1, and the text: On p. 9 it says baseline + 5 surveys, in Figure 1, however, there are seven survey waves in total. In Table 1 it says that the final survey was on 08/05/21, whereas Fig. 1 indicates surveys on 08/03/21 and 08/06/21. So perhaps you could streamline the figure with the text and table?

- Further, I noticed in Fig. 1 that the surveys were done under varying restrictions: Did you take a look whether it had an impact on your analyses if surveys were done under lockdown or eased restrictions?

- Perhaps you could also explain some more how exactly MET minutes are calculated? Is it all minutes > 1 MET? Since you used the IPAQ I assume that it’s only minutes > 3 MET? I.e., only minutes for vigorous or moderate physical activity or walking (measured by the IPAQ)? Further, is it simply a sum of minutes or is the intensity (i.e., the difference in MET for, e.g., walking vs. MVPA) somehow reflected in the score? For me, the term “MET minutes” is simply a bit confusing as all activities (i.e., also lying quietly) have a certain MET, which also depends on the individual, and it is probably difficult to assess all activities and associated METs based on survey data, so perhaps you could clarify the scores some more and discuss its limitations? Perhaps you could also give an example of what 1 MET minute represents?

- I am not quite sure about your dependent PA variable: Is it all PA surveys? I.e., baseline and the 5/6 follow-ups? And is it absolute PA values or changes in PA? From your methods and results section, I understand that it is absolute PA values (i.e., MET minutes per person per survey wave), in the Limitations (and Figure descriptions), however, you state that you measured changes in PA instead of absolute levels? Perhaps you could clarify this?

- Models: PA ~ loneliness + (1|participant) Is this the correct univariable model equation? I.e., there is no time variable indicating how much time has passed since baseline/loneliness assessment? Did you check whether time has an impact?

- How does the random intercept “allow[ed] for differing baseline (pre-pandemic) PA levels”? (p. 10) Are (retrospective) pre-pandemic PA levels also included in the univariable model?

- Missings of BMI: According to Table 2, BMI is missing for 66.7% of the participants, according to the text it’s 70.1%. Either way, I am wondering how missings impact Models 2 and 3. Perhaps you could include in S3 how many observations and participants were included in each model? Also, you might include the coefficients for the covariates? Then it might be easier to retrace model equations etc.

- I am also wondering whether you included loneliness and shielding in one model and took a look at the incremental effects?

Minor comments:

- 4.2: „Those who were „often lonely” at the onset of COVID-19 restrictions completed significantly less PA per week over the …”, “per” missing?

- 2.2: “whether the participant was living alone (yes no)”, p. 10 “/” missing

- P. 18: “Therefore, the negative health consequences of the pandemic will have longer term effects and impact beyond direct morbidity and mortality attributable to COVID-19” – perhaps be more careful and use “might”? After all, loneliness was only assessed at the beginning of the pandemic and loneliness and PA might have a bidirectional association or an association independent from the COVID pandemic?

- P 19. “However, these limitations are largely due to the fact that loneliness is subjective” “that” missing

- P. 19 “This may have been over- or under-estimated.” “these”?

Reviewer #2: The authors study a topic of high societal relevance and the strengths of the investigation include a large data set, however, the retrospective methods used limit the evidence and especially the statistics and its interpretation remain non-transparent to me.

Major comments:

1. The authors use multi level modelling, but it remains non-transparent whether they investigate a) within-person changes (e.g., PA changes from baseline to pandemic), or b) between-person changes (e.g., differences in PA between participants at baseline/or in pandemic conditions), or c) an interaction of both. This I could not detect neither from the methods/stats description nor the results section wording. To make this clear to readers I would suggests the authors to describe the multi level used in very detail, including the model equations. Moreover, I would suggest to carefully indicate whether this is focusing, within-, between-person change or both throughout the manuscript, also in the results and discussion parts.

2. The large sample size comes with high power and thus favors to significant findings. Therefore, I would suggest the authors to additionally report standardized effect sizes, and/or unstandardized ones, such as changes in MET minutes in relation to the MET intercept if all other predictors are kept zero. This can help to interpret practical relevance of findings. Related to this, absolute MET minutes per week should be included into the abstract, too.

3. BMI data was only available for about 1/4 of your sample. Co-variate models were non-significant. This was interpreted as a co-founding issue. Which role would you assign to the missing BMI data – May this be another interpretation for the non-significance?

4. Figures 2 and 3 appear to have the very similar content to me – did there potentially anything mix up?

5. Limitations, first point: One could also argue that this limitation is rather more critical when looking at changes compared to looking at absolute values since the range is smaller and low reliability matters more. I would suggest to rephrase.

Minor comments:

6. For readers not familiar with MET minutes I would suggest to introduce this parameter in details, e.g., stating current norms, recommendations etc..

7. For reliability of retrospective PA and loneliness questionnaires, I would suggest to at least include a detailed discussion how accelerometry and momentary social contact ratings on electronic diaries could make measurements more reliable. The following resources may guide this: https://www.sciencedirect.com/science/article/abs/pii/S146902921930809X?via%3Dihub;
https://pubmed.ncbi.nlm.nih.gov/32831643/

8. I would suggest to include a frequency table on the distribution of loneliness and contact ratings.

9. page 18: “the pandemic will have” – this is not proven, suggest to use “may”

6. PLOS authors have the option to publish the peer review history of their article (what does this mean?). If published, this will include your full peer review and any attached files.

Reviewer #1: No

Reviewer #2: No

---

## [Author Response · Author response to Decision Letter 0]

20 May 2023

Editor comments

1. More information should be provided on how MET was calculated

Thank you – we have now included this information in the Methods as per the comments below 

2. An explicit model description providing the actual lmer specifications is necessary for the readers

 to understand what was calculated

The model equations have now been included in supplementary materials S3

3. Your assessment of loneliness is rather unusual by asking one month in retrospect how participants rated their loneliness. A more common approach would be to use a dedicated questionnaire like the UCLA loneliness scale which provides a numeric and metric scale

We agree and raise the limitations of this approach in the discussion. The question used is derived from the Imperial College Sleep Questionnaire, but in its formulation aligns closely with that suggested by the Office for National Statistics with regards loneliness: ‘How often do you feel lonely?: Often/always, Some of the time, Occasionally, Hardly ever, Never’ (https://www.ons.gov.uk/peoplepopulationandcommunity/wellbeing/methodologies/measuringlonelinessguidanceforuseofthenationalindicatorsonsurveys#recommended-measures-for-adults).

4. Your questionnaire has ordinal properties as the difference between sometimes and never might not be equal compared to sometimes and often etc. This makes a linear model a strong assumption given that the predictor might not be linear and could represent an exponential increase in loneliness. The authors should consider using generalized linear mixed models with different specifications to check if treating loneliness as a linear predictor provides the best model fit. This kind of assessment should also be listed as a limitation. Furthermore, as reviewer 2 notes, high power also requires qualification of significant vs. meaningful. The authors could use Bayesian models using the brms and bayestestR packages to check for evidence in favor or against the null hypothesis in addition to frequentist statistics.

We have used linear regression models as our outcome (physical activity MET minutes) is a continuous variable. Loneliness is treated in the models as a categorical covariate. As shown in Figure 2, we have not treated loneliness as ordinal, as the difference between each ‘level’ within the category is not the same. We have now provided the model equations which hopefully clarifies the model specification. Regarding the use of a Bayesian model, we are unsure what the benefit of this approach would be in our study, given that there is no clear ‘prior’ on which to base our initial expectations from previous literature. We also believe that to add in an additional Bayesian approach could complicate the analysis and interpretation of results. With regards the clinical significance of the reductions in PA seen in those who were lonely or shielding, these have been incorporated into the discussion. The differences represent a significant proportion of weekly recommended PA as per WHO or UK CMO guidelines. 

Reviewer 1 major comments

1. There seems to be a mismatch of the number of surveys depicted in Fig. 1, Table 1, and the text: On p. 9 it says baseline + 5 surveys, in Figure 1, however, there are seven survey waves in total. In Table 1 it says that the final survey was on 08/05/21, whereas Fig. 1 indicates surveys on 08/03/21 and 08/06/21. So perhaps you could streamline the figure with the text and table?

Thank you for highlighting this, the figure has been updated to reflect the correct study dates. 

2. Further, I noticed in Fig. 1 that the surveys were done under varying restrictions: Did you take a look whether it had an impact on your analyses if surveys were done under lockdown or eased restrictions?

The analysis investigated PA over the entire time period. This is due to the fact that any change in physical activity would be delayed and so such short-term changes would not be seen. The first survey asked about pre pandemic PA; we referred to this previously as baseline PA but have amended as pre-pandemic PA to avoid confusion. We have also added this to the limitations for clarity:

‘Similarly, the surveys were completed under varying restrictions. Although adjustment for seasonality was performed, it may have been useful to investigate whether changes in restrictions themselves impacted on PA levels in the short term, which would not have been identifiable through the analysis presented here. The study is investigating the longer term changes in PA from COVID-19 restrictions, and it may also be useful to investigate the impact of external events, such as changes in social restrictions, on short-term changes in PA over the pandemic’

3. Perhaps you could also explain some more how exactly MET minutes are calculated? Is it all minutes > 1 MET? Since you used the IPAQ I assume that it’s only minutes > 3 MET? I.e., only minutes for vigorous or moderate physical activity or walking (measured by the IPAQ)? Further, is it simply a sum of minutes or is the intensity (i.e., the difference in MET for, e.g., walking vs. MVPA) somehow reflected in the score? For me, the term “MET minutes” is simply a bit confusing as all activities (i.e., also lying quietly) have a certain MET, which also depends on the individual, and it is probably difficult to assess all activities and associated METs based on survey data, so perhaps you could clarify the scores some more and discuss its limitations? Perhaps you could also give an example of what 1 MET minute represents?

Thank you for your comments on this. We have amended the text to provide further detail on how MET minutes were calculated and have provided an example of this. This has been added to the statistical methods section: 

‘IPAQ data were cleaned in accordance with IPAQ protocol (27). Weekly Metabolic Equivalent of Task (MET) minutes, which represent the number of minutes at a certain intensity of energy expenditure per week (as multiples of resting metabolic rate), were calculated for each participant at each survey wave, as well as for activity levels before the implementation of restrictions (pre-pandemic PA). MET minutes were calculated by multiplying the following MET score values as defined by the IPAQ protocol (walking = 3.3 METs, moderate PA = 4.0 METs, and vigorous PA = 8.0 METs) by the number of minutes completing the activity. For example, walking at a moderate pace for 5 minutes would represent 16.5 MET minutes (31).’

4. I am not quite sure about your dependent PA variable: Is it all PA surveys? I.e., baseline and the 5/6 follow-ups? And is it absolute PA values or changes in PA? From your methods and results section, I understand that it is absolute PA values (i.e., MET minutes per person per survey wave), in the Limitations (and Figure descriptions), however, you state that you measured changes in PA instead of absolute levels? Perhaps you could clarify this?

In the analysis we are looking at the change in PA across all surveys (from date 1 to date 2). When we say pre-pandemic this is before date 1 and this is adjusted for in all multivariable models. We are also looking at differences in PA between groups (i.e lonely vs not lonely) We have amended the text to make this clearer in the statistical methods section on page 10. 

‘The study will investigate between person differences in PA during the pandemic, adjusted for an individual’s pre-pandemic PA. Two-level univariable linear mixed models were used, incorporating random intercepts for each participant, to assess the associations between shielding status, and loneliness at the point of the first survey, and time varying PA.’

5. Models: PA ~ loneliness + (1|participant) Is this the correct univariable model equation? I.e., there is no time variable indicating how much time has passed since baseline/loneliness assessment? Did you check whether time has an impact?

In the analysis we do include a month variable. The aim of this is to account for time varying components and seasonality. Thank you for highlighting this, full model equations have been included in supplementary material S3.

6. How does the random intercept “allow[ed] for differing baseline (pre-pandemic) PA levels”? (p. 10) Are (retrospective) pre-pandemic PA levels also included in the univariable model?

We apologize the wording on random intercepts was confusing. We have changed this on page 10 in the statistical methods section. Pre-pandemic PA levels have not been included in the univariable model. 

7. Missings of BMI: According to Table 2, BMI is missing for 66.7% of the participants, according to the text it’s 70.1%. Either way, I am wondering how missings impact Models 2 and 3. Perhaps you could include in S3 how many observations and participants were included in each model? Also, you might include the coefficients for the covariates? Then it might be easier to retrace model equations etc.

Thank you for bringing this to our attention. The correct number has been clarified in the text (66.7%). Missingness around BMI has been clarified and discussed in the discussion. Tables of model observations and participants, as well as coefficients for covariates have been added in supplementary material S5. 

‘Another limitation of this study was the large amount of missing data for BMI. This is because many participants did not complete both the height and weight responses to the initial survey. This high level of missing data could have had implications for the results of this study, and affected the level of significance of results for both loneliness and shielding multivariable models.’ 

8. I am also wondering whether you included loneliness and shielding in one model and took a look at the incremental effects?

We decided not to include both variables within the same model due to the fact that we could be adjusting for variables on the causal pathway. We think this would be interesting to look at; however it is not within the scope of our study and would need causal mediation analysis.

Minor Comments 

1. 4.2: „Those who were „often lonely” at the onset of COVID-19 restrictions completed significantly less PA per week over the …”, “per” missing?

Thank you for bringing this to our attention, this has been amended in the text. 

2. 2.2: “whether the participant was living alone (yes no)”, p. 10 “/” missing

Thank you for bringing this to our attention, this has been amended in the text. 

3. P. 18: “Therefore, the negative health consequences of the pandemic will have longer term effects and impact beyond direct morbidity and mortality attributable to COVID-19” – perhaps be more careful and use “might”? After all, loneliness was only assessed at the beginning of the pandemic and loneliness and PA might have a bidirectional association or an association independent from the COVID pandemic?

Thank you for bringing this to our attention, this has been amended in the text. 

4. P 19. “However, these limitations are largely due to the fact that loneliness is subjective” “that” missing

Thank you for bringing this to our attention, this has been amended in the text. 

5. P. 19 “This may have been over- or under-estimated.” “these”?

Thank you for bringing this to our attention, this has been amended in the text. 

Reviewer 2 Major Comments

1. The authors use multi level modelling, but it remains non-transparent whether they investigate a) within-person changes (e.g., PA changes from baseline to pandemic), or b) between-person changes (e.g., differences in PA between participants at baseline/or in pandemic conditions), or c) an interaction of both. This I could not detect neither from the methods/stats description nor the results section wording. To make this clear to readers I would suggests the authors to describe the multi level used in very detail, including the model equations. Moreover, I would suggest to carefully indicate whether this is focusing, within-, between-person change or both throughout the manuscript, also in the results and discussion parts.

To clarify our methodology and study question - we are looking at between person differences in PA during the pandemic, adjusted for individuals pre pandemic activity. This has been reflected on page 10 in the statistical methods section. The model equations have been included in S3 and referenced in the methods section of the text. 

Univariable: 

yi,j= β0+ β 1Z+ µi

 Model 1:

yi,j= β0+B1Zj+ β2age,j+ β3sex,j+ β4ethnicity,j+ β5Monthi.j + β6PAprepandemic,j + µi

 Model 2:

yi,j= β0+ β1Zj+ β2age,j+ β3sex,j+ β4ethnicity,j+ β5Monthi.j+ β6UnderlyingConditions,j+ β7BMI,j+ β7PAprepandemic,j + µi

 Model 3:

yi,j= β0+ β 1Zj+ β2age,j+ β3sex,j+ β4ethnicity,j+ β5Monthi.j+ β6UnderlyingConditions,j+ β7BMI,j+ β8AlcoholDrinker,j+ β9Smoker,j+ β10LivingAlone+ β11RelationshipStatus+ β12PAprepandemic,j + µi

Where i is survey wave in participant j, Z is the exposure, and y is PA 

2. The large sample size comes with high power and thus favors to significant findings. Therefore, I would suggest the authors to additionally report standardized effect sizes, and/or unstandardized ones, such as changes in MET minutes in relation to the MET intercept if all other predictors are kept zero. This can help to interpret practical relevance of findings. Related to this, absolute MET minutes per week should be included into the abstract, too.

Our focus of this study is to look at the predictors of physical activity over time during the pandemic. There is no standard population by which to report standardised effect sizes. Physical inactivity is a public health issue concerning people of all ages across society. The focus of the study is to look at inactivity among vulnerable groups. Our concern, if we set all predictors to 0, would be that we are providing an estimate only for the most prevalent groups which may not provide a representative view of the study population. 

3. BMI data was only available for about 1/4 of your sample. Co-variate models were non-significant. This was interpreted as a co-founding issue. Which role would you assign to the missing BMI data – May this be another interpretation for the non-significance?

Thank you for highlighting this issue. We have amended the text to discuss BMI missing data more thoroughly in the limitations section.

‘Another limitation of this study was the large amount of missing data for BMI. This is because many participants did not complete both the height and weight responses to the initial survey. This high level of missing data could have had implications for the results of this study, and affected the level of significance of results for both loneliness and shielding multivariable models.’ 

4. Figures 2 and 3 appear to have the very similar content to me – did there potentially anything mix up?

Thank you for highlighting this, this has now been amended. 

5. Limitations, first point: One could also argue that this limitation is rather more critical when looking at changes compared to looking at absolute values since the range is smaller and low reliability matters more. I would suggest to rephrase.

Thank you for raising this point. The text has been amended to discuss this more thoroughly in the limitations section. We have removed sentence referencing change in PA. If we are looking at differences over time systematic over/ under reporting will be less of an issue. 

Reviewer 2 Minor Comments 

1. For readers not familiar with MET minutes I would suggest to introduce this parameter in details, e.g., stating current norms, recommendations etc.. 

The text has been amended to describe MET min more thoroughly:

‘IPAQ data were cleaned in accordance with IPAQ protocol (27). Weekly Metabolic Equivalent of Task (MET) minutes, which represent the number of minutes at a certain intensity of energy expenditure per week (as multiples of resting metabolic rate), were calculated for each participant at each survey wave, as well as for activity levels before the implementation of restrictions (pre-pandemic PA). MET minutes were calculated by multiplying the following MET score values as defined by the IPAQ protocol (walking = 3.3 METs, moderate PA = 4.0 METs, and vigorous PA = 8.0 METs) by the number of minutes completing the activity. For example, walking at a moderate pace for 5 minutes would represent 16.5 MET minutes (31)’.

2. For reliability of retrospective PA and loneliness questionnaires, I would suggest to at least include a detailed discussion how accelerometry and momentary social contact ratings on electronic diaries could make measurements more reliable. The following resources may guide this: https://www.sciencedirect.com/science/article/abs/pii/S146902921930809X?via%3Dihub;
https://pubmed.ncbi.nlm.nih.gov/32831643/

Thank you for highlighting this. A section has been added in the discussion on page 19 exploring how accelerometry and momentary social contact ratings on electronic diaries could make measurements more reliable. 

3. I would suggest to include a frequency table on the distribution of loneliness and contact ratings.

The predictors of loneliness and social isolation were taken at the point of the first (pre-pandemic) survey, this is shown in table 2. 

4. page 18: “the pandemic will have” – this is not proven, suggest to use “may”

Thank you for brining this to our attention, this has been amended in the text.

---

## [Decision Letter · Decision Letter 1]

2 Aug 2023

The impacts of social restrictions during the COVID-19 pandemic on the physical activity levels of over 50-year olds: the CHARIOT COVID-19 Rapid Response (CCRR) cohort study

PONE-D-23-02121R1

Dear Dr. Salman,

We’re pleased to inform you that your manuscript has been judged scientifically suitable for publication and will be formally accepted for publication once it meets all outstanding technical requirements.

Kind regards,

Julian Packheiser

Academic Editor

PLOS ONE

Additional Editor Comments (optional):

Reviewers' comments:

Reviewer's Responses to Questions

**Comments to the Author**

1. If the authors have adequately addressed your comments raised in a previous round of review and you feel that this manuscript is now acceptable for publication, you may indicate that here to bypass the “Comments to the Author” section, enter your conflict of interest statement in the “Confidential to Editor” section, and submit your "Accept" recommendation.

Reviewer #1: All comments have been addressed

2. Is the manuscript technically sound, and do the data support the conclusions?

Reviewer #1: Yes

3. Has the statistical analysis been performed appropriately and rigorously? 

Reviewer #1: Yes

4. Have the authors made all data underlying the findings in their manuscript fully available?

Reviewer #1: No

5. Is the manuscript presented in an intelligible fashion and written in standard English?

Reviewer #1: Yes

6. Review Comments to the Author

Reviewer #1: (No Response)

7. PLOS authors have the option to publish the peer review history of their article (what does this mean?). If published, this will include your full peer review and any attached files.

Reviewer #1: No

---

## [Editor Report · Acceptance letter]

15 Sep 2023

PONE-D-23-02121R1 

The impacts of social restrictions during the COVID-19 pandemic on the physical activity levels of over 50-year olds: the CHARIOT COVID-19 Rapid Response (CCRR) cohort study 

Dear Dr. Salman:

I'm pleased to inform you that your manuscript has been deemed suitable for publication in PLOS ONE. Congratulations! Your manuscript is now with our production department. 

Kind regards, 

on behalf of

Dr. Julian Packheiser 

Academic Editor

PLOS ONE